# Detection of A and B Influenza Viruses by Surface-Enhanced Raman Scattering Spectroscopy and Machine Learning

**DOI:** 10.3390/bios12121065

**Published:** 2022-11-23

**Authors:** Artem Tabarov, Vladimir Vitkin, Olga Andreeva, Arina Shemanaeva, Evgeniy Popov, Alexander Dobroslavin, Valeria Kurikova, Olga Kuznetsova, Konstantin Grigorenko, Ivan Tzibizov, Anton Kovalev, Vitaliy Savchenko, Alyona Zheltuhina, Andrey Gorshkov, Daria Danilenko

**Affiliations:** 1Institute of Advanced Data Transfer Systems, ITMO University, Birzhevaya Liniya 14, 199034 Saint Petersburg, Russia; 2National Research Center “Kurchatov Institute”, Akademika Kurchatova Sq. 1, 123182 Moscow, Russia; 3Smorodintsev Research Institute of Influenza, Prof. Popova Str. 15/17, 197376 Saint Petersburg, Russia

**Keywords:** surface-enhanced Raman spectroscopy, SERS, influenza A virus, influenza B virus, detection, machine learning

## Abstract

We demonstrate the possibility of applying surface-enhanced Raman spectroscopy (SERS) combined with machine learning technology to detect and differentiate influenza type A and B viruses in a buffer environment. The SERS spectra of the influenza viruses do not possess specific peaks that allow for their straight classification and detection. Machine learning technologies (particularly, the support vector machine method) enabled the differentiation of samples containing influenza A and B viruses using SERS with an accuracy of 93% at a concentration of 200 μg/mL. The minimum detectable concentration of the virus in the sample using the proposed approach was ~0.05 μg/mL of protein (according to the Lowry protein assay), and the detection accuracy of a sample with this pathogen concentration was 84%.

## 1. Introduction

Outbreaks and epidemics of viral diseases in recent years have raised the active search for new methods for differential diagnostics and virus detection. Modern methods of virus detection, such as polymerase chain reactions (PCR) [1] and enzyme-linked immunosorbent assays (ELISA) [2], have high sensitivity in determining the presence of influenza viruses in a sample [3,4] but have a number of disadvantages: laboriousness, time duration of the assays, low versatility, and a high percentage of false positive results [5,6].

Surface-enhanced Raman spectroscopy (SERS) is considered a promising alternative for the differential detection of viral infections [7,8,9,10,11]. The viral envelope contains surface proteins that differ in their amino acid composition and conformation which leads to the difference in Raman spectra of various viruses. Since the Raman scattering signal is rather weak, the SERS is used to enhance it by implementing special substrates with a surface layer containing nanoparticles. The latter allows amplification of the Raman scattering signal of the test sample due to the plasmon resonance effect, by 10^3^–10^9^ times, and the subsequent detection of the pathogen in low concentrations (5 × 10^8^ PFU/mL) [7]. Identification of viral particles by SERS spectroscopy has a number of potential advantages over classical diagnostic methods, such as the speed of testing and the possibility of diagnosing without the need to use specific antibodies or aptamers [7,9,10,11]. On the other hand, there are developments that are aimed at the selective detection of a specific pathogen based on SERS substrates made from precious metal nanoparticles and immobilized antibodies or aptamers [12,13,14]. This approach, however, deprives versatility, which is the SERS technology’s main advantage, since each antigen has its own specific features in the spectrum [15]. The label-free SERS platform, in combination with machine learning, can become a powerful tool for diagnosing viral diseases not only due to antigen detection, but also due to the detection of biomarkers in the sample [16,17].

Many research efforts are underway to create a universal SERS substrate made from metal nanoparticles and nanorods which will allow us to determine the presence of viruses (including their type) in a sample [18,19,20,21,22]. The optimal parameters of nanostructures in these studies are selected to amplify the Raman signal. Thus, the differences in spectra could be noticeable visually [22] or be differentiated by classical methods of mathematical analysis, such as multivariate calibration and partial least squares regression (PLS/PSR) [23]. It should be noted, however, that these studies were carried out with purified samples of viral particles. In field work dealing with clinical samples, virus detection using this approach can be complicated.

Spectral patterns of different influenza strains have similarities [24,25] and to make their robust classification possible, mathematical processing methods can be used. For example, machine learning technologies improve the accuracy of differentiation and classification of SERS spectra [26]. Huang J. et al. managed to achieve an accuracy of 87.7% using the technology of recurrent neural networks (RNNs) in identifying the spectra of SARS-CoV-2 spike proteins [27]. The experiments by Paria D. et al. demonstrated how the random forest algorithm provides an accuracy of 83% when differentiating Zika, coronavirus (SARS-CoV-2), influenza A (H1N1), and Marburg viruses [11]. Yeh Y.T. et al. described a technique for recognizing differences in the SERS spectra of rhinovirus, the influenza A virus, and the type 3 parainfluenza virus with an accuracy of 93% [24]. Their scientific team also conducted a study on samples containing avian influenza viruses and confirmed the data processing efficiency by machine learning using the logistic regression algorithm [24,28]. The work of Lim J.Y. et al. [25] demonstrates the possibility of classifying the cells infected with wild and mutant influenza A viruses by their SERS spectra using the principal component analysis (PCA) [29].

The paper by Zhang Z. et al. described an experiment on modifying the structures of silver nanosubstrates by adding acetonitrile, bromine, and calcium ions; the minimum number of viruses for successful detection, in this case, was 100 particles per test [30]. In their work, the PCA was used to classify the samples containing and not containing viruses; as a result, differentiation analysis showed the spectra separation with a 95% accuracy (yet the researchers noted that the cell proteins interfere with the detection of the viral proteins themselves). In continuing experiments with SERS substrates, Zhang Z. et al. used sodium borohydride to change the conformation of silver nanostructures in order to detect adenovirus, coronavirus (SARS-CoV-2), and the influenza A (H1N1) virus [31]. The analysis time was 2 minutes and the minimum number of viral particles for detection was reduced to 10 units. Their work used the Latent Dirichlet allocation algorithm [32] with a differentiation accuracy of 95%.

The work of Durmanov et al. describes the usage of a fabricated SERS substrate, which was composed of nanoporous mica with the addition of a thin silver layer by electron beam physical vapor deposition method [33]. Four types of viruses were selected to demonstrate the practical application of the new SERS substrate: the myxoma virus (MYXV), the canine distemper virus (CDV), the tobacco mosaic virus (TMV), and the potato virus X (PVX). The SERS substrate performance was tested in its ability to obtain the spectra of viral particles of different sizes, morphology, structural composition, and physicochemical properties. Various methods of data analysis were used to identify the viruses by their spectra. Data spectral analyses were carried out by the method of machine learning, the principal components, and the linear discriminant analysis (PCA-LDA), in particular. The classification model accuracy was double-checked by 5-fold repetition of the 3-fold cross-validation, resulting in an average accuracy of 99.4%.

Paria D. et al. reported on the creation of a label-free SERS platform with a metal-insulator-metal nanostructure [34]. Combined with machine learning methods, this structure made it possible to differentiate four viruses: the influenza A virus, coronavirus, Zika virus, and Marburg virus. The PCA method was used for visual analysis of the data obtained from the SERS spectra collection, and the random forest algorithm was used to classify the spectral data set. The resulting classification accuracy for labeling an unknown virus sample was ranging from 83 to 95%.

Song C. et al. were able to demonstrate the possibility of using the portable Raman spectrometer to detect the influenza A virus [35]. In their work, SERS spectra of the three influenza virus types (A/Mute Swan/MI/06/451072-2/2006, A/chicken/Pennsylvania/13609/1993, and A/chicken/TX/167280-4/02) and the control sample spectra were obtained, with the PCA method used to classify the spectra. Visualization of the results on the principal component axes indicated 100% accuracy (n = 10 for each sample). Despite the small number of data set samples, this study demonstrated the portability and versatility of the SERS virus particle detection technology.

The support vector machine (SVM) method has been previously applied to the detection of varied biomarkers using spectroscopic data [36]. A recent study by Yang Y. et al. demonstrates highly accurate differentiation of respiratory disease virus agents by implementing custom-fabricated SERS substrates with silver nanorods and the SVM classification procedure with data preprocessing [37]. The authors were able to differentiate a variety of viruses, including a coronavirus, influenza A and B viruses, and adenovirus. In their work, influenza viruses were contained in chick embryo allantoic fluid that could influence spectral uniformity and produce fluorescence. The inclusion of organic components may influence pattern recognition and defining fingerprints characteristic for viral particles.

In this paper, we propose implementing SERS and subsequent processing of the spectra to detect and differentiate the A and B influenza viruses in an STE buffer medium and investigate the limits of virus detection using this approach. Here, to the best of our knowledge, we, for the first time, demonstrate differentiating the A and B influenza viruses in an STE buffer based on SERS and SVM. We demonstrate successful detection and 93% accuracy in the differentiation of the viruses with low-cost commercial substrates, a simple STE buffer solution, and a standard machine learning algorithm that does not require time-consuming preprocessing steps of the spectral data. This makes our approach affordable and effective for use in real-life applications.

## 2. Materials and Methods

### 2.1. Viruses

Influenza A and influenza B viruses are typical agents causing acute respiratory infections. The antigenic structure of the hemagglutinin of influenza A and B viruses differs by ~70% [38], which made these pathogens a representative model for research.

Influenza A (A/California/07/2009, A(H1N1)pdm09) and influenza B (B/Hong Kong/269/2017) viruses were grown in chick embryos; purified virus concentrate was then obtained by differential centrifugation of virus-containing allantoic fluid followed by ultracentrifugation reprecipitation. The Pierce BCA Protein Assay Kit (Thermo Fisher Scientific, Rockford, IL, USA) was used to measure the protein concentration in purified virus concentrates. The result of determining the concentration of protein, according to Lowry, in concentrated viral suspensions, was <200 µg/mL. The hemagglutination titer of the concentrate was 1:256. The specificity of the obtained viral concentrates was confirmed by the ELISA test using specific monoclonal antibodies to the hemagglutinating protein of influenza A and B viruses.

The suspension of purified viral particles was placed in an STE buffer medium for further storage and use.

### 2.2. Buffer Medium

The STE containing NaCl, Tris, and EDTA with pH = 8.0 was used as a buffer medium. The choice of this medium was justified by the absence of a fluorescence signal in the range required. The absorption spectrum of the STE buffer medium (in the visible and IR ranges) is shown in Figure 1. It absorbs, in the IR region, near 980 nm lying far from the SERS signal.

### 2.3. SERS Substrates

Commercial SERS substrates from SERSitive [39] were used in this work. This choice was made due to the substrate’s hydrophilicity and tenfold Raman signal amplification at a 633 nm wavelength. The substrates consisted of glass, indium tin oxide, and contained nanostructured electrodeposited silver nanoparticles. The average size of substrate nanoparticles was 100–150 nm and the distance between nanoparticles was 100–200 nm. This substrate structure makes it possible to adsorb viral particles and provide high signal reproducibility over the entire surface. The image of the substrate obtained by scanning electron microscopy (the accelerating voltage was 10 kV) is shown in Figure 2.

### 2.4. Raman Spectroscopy Setup

The measurements were carried out using a Horiba LabRam Raman spectrometer (Horiba Jobin Yvon S.A.S., Longjumeau, France). The setup included a laser source (a 632.8 nm wavelength, 0.01 mW of power), a spectrometer (600 lines/mm grating, a 500 nm blaze wavelength), a Mitutoyo Apo Plan 50x VIS lens, and a CCD camera (a 2000 × 800-pixel matrix in the receiving area, a 15 × 15 μm size of the pixel). The design of the spectrometer is shown in Figure 3.

The excitation radiation of the laser source was put through an aperture diaphragm serving as a spatial filter. The edge filter reflecting the excitation light and transmitting the Raman signal directs the excitation beam to the sample fixed at the object stage. The Raman scattering signal was then collected by the lens, passed through the edge filter, and directed to the spectrometer. The spectrally resolved signal was projected onto a CCD camera to obtain a Raman spectrum.

It was experimentally found that the selected laser’s power and wavelength make possible the fluorescence signal reduction and allow for minimizing the probability of destroying the biological structures due to the nanoparticle heating.

## 3. Results

### 3.1. Spectra of A Pure Buffer Medium and Viruses in A Buffer Medium

During the experiments, the first dark spectra and spectra of a pure substrate without a sample were obtained, with no Raman scattering peaks or artifacts visible during the process. Then, 1 µL volume samples of a clean STE buffer medium with influenza A and B viral particles (having concentrations of 500 µg/mL and 200 µg/mL according to the Lowry method, correspondingly), were applied to the SERS substrates surface. Further, the samples were dried at room temperature and placed in the spectrometer. SERS spectra were obtained in the range from 550 to 2000 cm^−1^, which includes the main vibrational modes of organic compounds; the exposure time was 60 s. Several spectra were taken from five to six randomly selected spatial points on the substrate. As a result, 25–30 spectra were obtained for each sample, some of which are given in Figure 4.

The Raman spectra of the influenza A and B containing samples are visually similar to the buffer medium spectra due to the strong signal from the latter. In this case, the differentiation of spectra and virus detection become complicated (see Figure 5), prompting the use of mathematical processing methods (see Section 3.3).

### 3.2. Spectra of Influenza A Virus in A Buffer Medium at Different Concentrations

To determine the method sensitivity limits of the SERS spectra and classification using mathematical processing, the suspension with the influenza A virus was diluted with the STE buffer medium in various ratios: 1:10, 1:100, 1:1000, and 1:10,000. The initial protein concentration of the sample with the viral suspension was 500 µg/mL, according to the Lowry method [40]. For this purpose, the STE buffer medium was poured into four test tubes in portions of 10 μL. Then, 1 µL of the influenza A virus concentrated suspension was added to the first tube and thoroughly mixed. Next, 1 µL was taken from the resulting suspension and put into the next tube, etc. The protein concentration in the resulting diluted samples was 50 μg/mL, 5 μg/mL, 0.5 μg/mL, and 0.05 μg/mL, respectively. The samples were then placed on a SERS substrate and dried. The process of obtaining Raman scattering spectra was the same as the method described in Section 3.1; ten samples were obtained for each concentration. The spectra did not differ visually, as shown in Figure 6.

### 3.3. Mathematical Processing of Spectra

#### 3.3.1. Detection of Influenza A Virus in a Buffer Medium

The experimental spectra obtained were combined into a table in which the columns corresponded to wavelengths and the rows corresponded to intensity values. The table that is used as a sample set had the dimensions of 57 × 1332 cells, where 29 lines corresponded to the spectra of the STE buffer medium and 28 lines corresponded to the spectra of the influenza A virus. For the mathematical processing, we used the package Scikit-learn v. 0.24.2 for Python v. 3.10.7 [41].

The detection of the viral particles was carried out by binarily classifying the spectra of the pure buffer medium and the spectra of virus A in a buffer medium (with a concentration of 500 µg/mL, according to Lowry). For preliminary processing of the data, we applied standardization (StandardScaler) and normalization (Normalizer) to the sample set. For classification, the support vector machine (SVM) method was used. The SVM is one of the most commonly used machine learning algorithms, the purpose of which is to solve the classification problem by constructing an optimal separating hyperplane [42].

Samples that are closest to the hyperplane are the support vectors. The hyperplane is constructed so that the distance between it and the support vectors is maximized. This distance is called a margin. Accordingly, the remaining objects must have a distance to the hyperplane greater than the margin to perform the classification. This strict rule corresponds to the linear kernel.

Cross-validation was carried out using StratifiedKFold (stratification of the data set into two subsets at a ratio of 9 to 1, a training set of 51–52 samples, and a test set of 5–6 samples, with equal distribution of objects belonging to the different classes).

The average virus detection accuracy was 95.5% (see Table 1). Visualization of the classification is shown in Figure 7 (in this case, the values of the spectra were separated by a hyperplane).

#### 3.3.2. Differentiation of Influenza A, Influenza B, and Pure Buffer Medium Spectra

The spectra of influenza A and B viruses were combined into a table and the total size was 90 × 1332 cells (a total of 90 spectrum samples). An additional class-labeled column was introduced for classification purposes (the blank buffer medium was labeled 0, the influenza A virus was labeled 1, and the influenza B virus was labeled 2). Each class of spectra was represented by an equal number of samples (30 for each). Similarly, the methods of standardization (StandardScaler) and normalization (Normalizer) of the samples were used for preprocessing. Here, we used the SVM method with the hinge loss kernel function for classification that introduces soft boundaries of a margin.

Cross-validation was carried out using the StratifiedKFold method (the total data set was stratified into two subsets at a ratio of 9 to 1, where the size of the training set was 81 and the size of the test set was 9 samples. The training (test) set included per 27 (3) spectrum samples of the virus A, the virus B, and the pure buffer medium. Thus, a balanced division of the sample set into subsets with respect to the classes was obtained). In the full dataset, the mean cross-validation accuracy of classification was 93% (Table 2). For different subsamples, the prediction accuracy varied within 85–100% and the standard deviation (SD) was 4.8% (0.04835) at 10 cross-validation iterations. The visualization results are shown in Figure 8.

#### 3.3.3. Determination of the Minimum Allowable Concentration of Viral Particles for The Detection and Classification of the Influenza A Virus

In order to solve the binary classification problem of the pure buffer medium spectra from the influenza A virus spectra at different dilutions, the support vector machine with a linear classification kernel was used.

For samples with different influenza A virus concentrations in the buffer medium (1:10, 1:100, 1:1000, and 1:10,000), a data set was formed, which consisted of 10 measurements of the virus A spectra (for each concentration) and 10 measurements of the buffer spectra. Five subsets were formed for cross-validation (cross-validation was carried out using the stratification method to maintain the balance of virus and buffer classes). Thus, the test set for cross-validation had a dimension of 4 × 1332 (two spectra of the influenza A virus and two spectra of the pure buffer medium), and the training set was 16 × 1332 (eight spectra of the influenza A virus and eight spectra of the pure buffer medium).

The model always accurately differentiated the virus and the buffer at high virus concentrations (the classification accuracy was 100%), but the classification accuracy decreased while lowering the concentration. The results of differentiation are presented in Table 3. The samples with a virus concentration of 1:10,000 are of the greatest practical interest, and the average accuracy of the virus/buffer binary classification for the test sets was 0.84 (84%) in this case.

We determined the spectrum relative SD (RSD) to investigate the intensity signal deviation of the spectrum samples and their potential influence on the classification accuracy. We calculated the normalized intensity SD at each spectral wavenumber, normalized it by a mean value to obtain an RSD at each point, and then averaged it over the entire wavenumber range. The average RSD was 24% for 1:10, 42% for 1:100, 26% for 1:100, and 28% for 1:10,000. It should be noted that such relatively high values may be caused by using a label-free SERS substrate, and that no special techniques for ensuring uniformity of a sample distribution were undertaken. Despite these RSD values, no apparent influence of signal deviation was found during the classification process.

## 4. Discussion

Through a combination with machine learning, photonics technology has flourished in biosensing by the implementation of extremely sophisticated methods to prepare biosamples, tailored substrates, and complicated preprocessing of spectral data for numerical analysis. In this work, we were focusing on design approaches to easy-to-use and affordable biosensing applications, and demonstrated the possibility of detecting and differentiating viral particles in a sample containing the STE buffer medium. The use of surface-enhanced Raman spectroscopy technology in combination with machine learning algorithms made it possible to differentiate samples with the influenza A virus, the influenza B virus, and without the virus, even for the limited data set sizes. The spectra were classified using the support vector machine. The accuracy of virus detection was 93%. To approach clinical studies, the concentration of the influenza A virus was reduced by 10^4^ times. At low concentrations, the method used was able to differentiate between virus-free and virus-containing samples with an accuracy of 84%. The results of the study will help to develop fast, cheap but reliable diagnostic methods for real life working with clinical samples.

## Figures and Tables

**Figure 1 biosensors-12-01065-f001:**
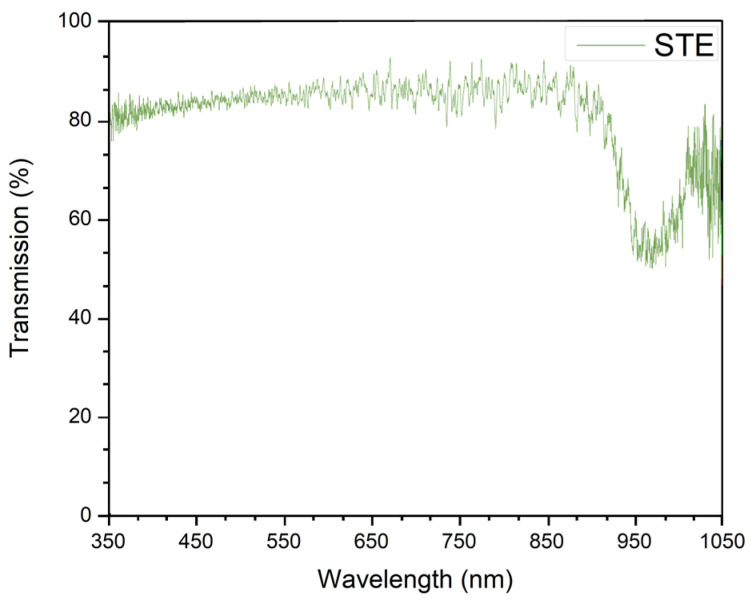
Absorption spectra of the STE buffer medium.

**Figure 2 biosensors-12-01065-f002:**
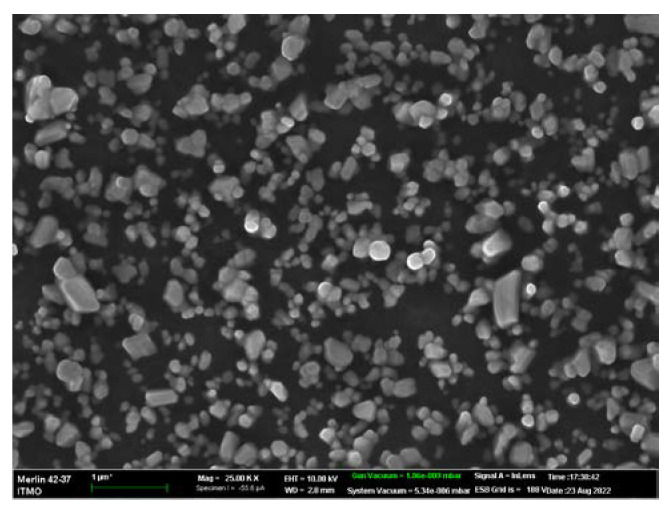
Scanning electron microscopy image of SERS substrate.

**Figure 3 biosensors-12-01065-f003:**
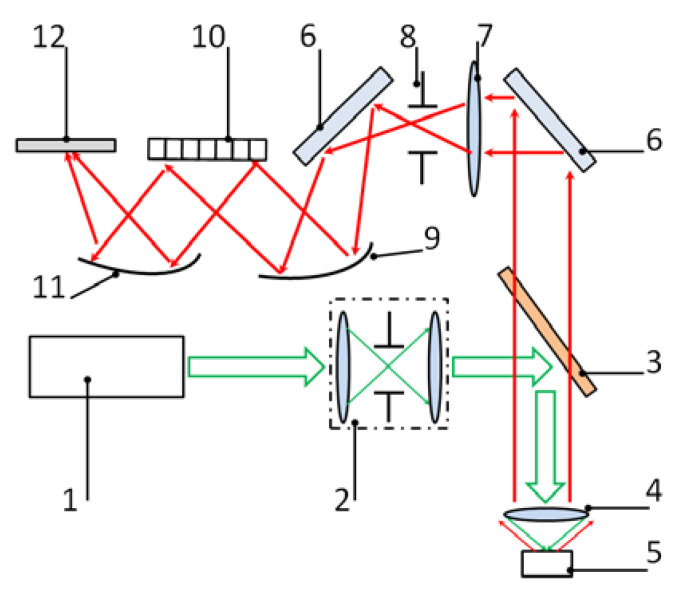
Design of the Raman spectroscopy setup: 1—laser source; 2—spatial filter; 3—edge-filter; 4—lens; 5—sample; 6—mirror; 7—condenser; 8—entrance slit of the spectrometer; 9, 11—aspherical mirrors; 10—diffraction grating; 12—CCD camera.

**Figure 4 biosensors-12-01065-f004:**
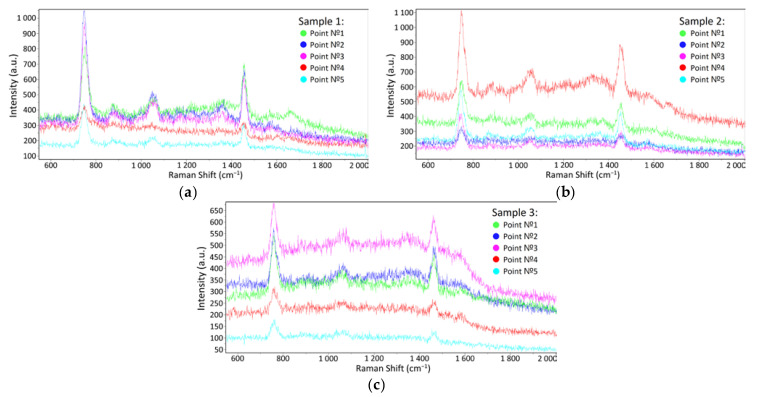
SERS spectra examples at different spatial points on the substrate having: (**a**) a pure buffer medium; (**b**) an influenza A virus in the buffer medium; (**c**) an influenza B virus in the buffer medium. The spectra of one sample taken at different points are indicated by different line colors.

**Figure 5 biosensors-12-01065-f005:**
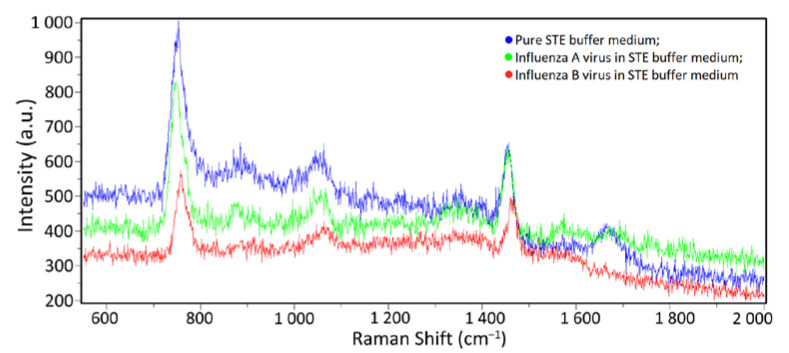
Relative comparison of the spectra: blue line—the pure STE buffer medium; green line—influenza A virus in the STE buffer medium; red line—influenza B virus in the STE buffer medium.

**Figure 6 biosensors-12-01065-f006:**
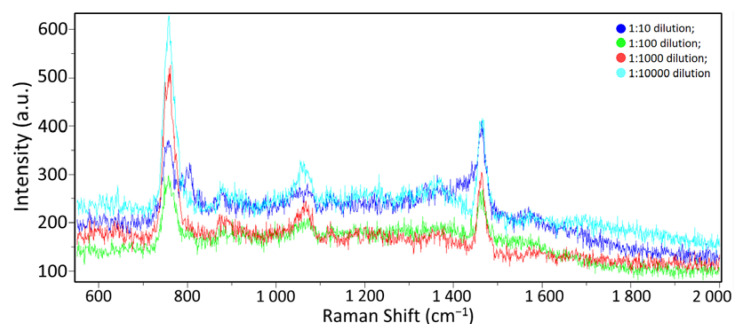
Influenza A virus in different dilutions: dark blue line—1:10 dilution; green line—1:100 dilution; red line—1:1000 dilution; light blue line—1:10,000 dilution.

**Figure 7 biosensors-12-01065-f007:**
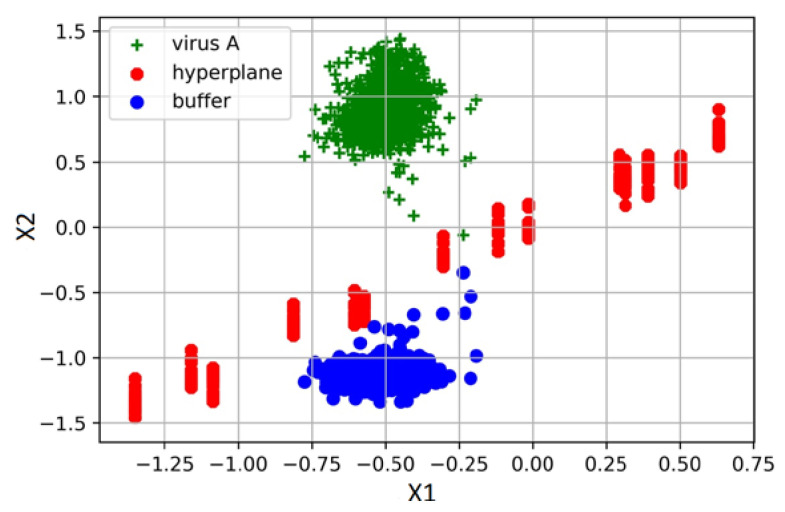
Visualization of spectra clustering of the pure STE buffer medium and influenza A virus in the STE buffer medium using projection onto support vectors X1 and X2. The red dots mark the projection of hyperplane points, which allows to classification of the spectra.

**Figure 8 biosensors-12-01065-f008:**
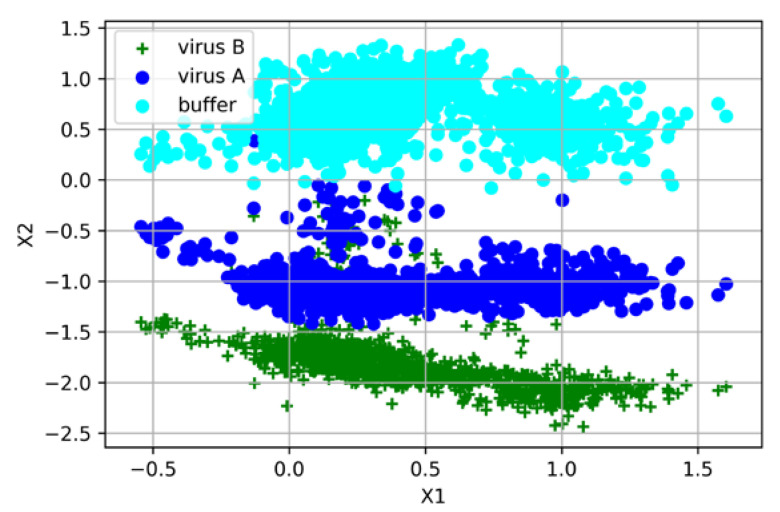
Visual representation of the spectra classification by support vector machine showing the projection of the samples onto support vectors X1 and X2: light blue—the pure STE buffer medium; dark blue—the influenza A virus in the STE buffer medium; green—the influenza B virus in the STE buffer medium.

**Table 1 biosensors-12-01065-t001:** Spectrum Classification Accuracy Results.

Iteration	1	2	3	4	5	6	7	8	9	10
Test set size	6	6	6	6	6	6	6	5	5	5
Accuracy	100%	94%	94%	100%	94%	94%	94%	88%	100%	94%

**Table 2 biosensors-12-01065-t002:** Accuracy of the pure STE buffer medium, the influenza A virus in the STE buffer medium, and the influenza B virus in the STE buffer medium spectrum differentiation.

Iteration	1	2	3	4	5	6	7	8	9	10
Accuracy	93%	89%	93%	100%	93%	96%	89%	85%	100%	96%
SD	4.8%									

**Table 3 biosensors-12-01065-t003:** Results of spectra differentiation by the algorithm at different influenza A virus dilutions.

Sample Dilution	Average Accuracy	SD of Accuracy	Average Spectrum RSD	Total Number of Samples	Training Set Size	Test Set Size
1:10	100%	0%	24%	20	15	5
1:100	100%	0%	42%	20	15	5
1:1000	94.5%	6.9%	26%	24	20	4
1:10,000	84%	15.2%	28%	20	15	5

## Data Availability

The data presented in this study are available on reasonable request from the corresponding authors.

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
