# Peer review of "Detection of A and B Influenza Viruses by Surface-Enhanced Raman Scattering Spectroscopy and Machine Learning"

_biosensors, 2022, doi:10.3390/bios12121065_

Round 1
Reviewer 1 Report
The authors presented a study on the detection of A and B Influenza viruses by machine learning-supported SERS spectroscopy with the use of the support vector machine to process the SERS spectra.
The study is well-organized, thoroughly conducted, and convincingly presented. However, there are a number of publications showing a similar approach to virus detection e.g.
S. Modak et al. Application of Support Vector Machines in Viral Biology, Global Virology III: Virology in the 21st Century, Springer Nature, (2019) ISBN : 978-3-030-29021-4
The author should clearly describe what is new in their approach with respect to the existing literature presenting the same path.
Lesser comments refer to:
· The authors stated that the used substrates provide high signal reproducibility over the entire surface. Can the authors provide the RSD of the signal for the detection of the influenza virus? How large RSD might influence the results retrieved by the used algorithm?
· Figure 4 is of poor quality. Please increase the fonts on the axes.
Author Response
We would like to thank all the reviewers for giving their time and energy in reviewing our manuscript. We believe that their critiques have greatly improved the quality and accessibility of our paper. It is now much more self-contained regarding the approach as a lot more details and references has been included in the revised manuscript.
Reviewer 1
The authors presented a study on the detection of A and B Influenza viruses by machine learning-supported SERS spectroscopy with the use of the support vector machine to process the SERS spectra.
The study is well-organized, thoroughly conducted, and convincingly presented.
We thank the reviewer for their kind words.
However, there are a number of publications showing a similar approach to virus detection e.g.
S. Modak et al. Application of Support Vector Machines in Viral Biology, Global Virology III: Virology in the 21st Century, Springer Nature, (2019) ISBN : 978-3-030-29021-4
The author should clearly describe what is new in their approach with respect to the existing literature presenting the same path.
We agree with the reviewer and have significantly increased the details related to the key aspects of our work. We believe that this has significantly improved the manuscript.
Our work demonstrates successful detection and differentiation of the viruses with low-cost commercial substrates, simple STE buffer solution, and standard machine learning algorithm which does not require time-consuming preprocessing steps. We do not exploit complicated preprocessing of the spectral data and expensive tailored substrates, but still demonstrate 93% accuracy in differentiation of the viruses what makes our approach more affordable and effective for use in the real life applications.
Меньшие комментарии относятся к:
- Авторы заявили, что используемые подложки обеспечивают высокую воспроизводимость сигнала по всей поверхности. Могут ли авторы предоставить ОСО сигнала для обнаружения вируса гриппа?
Мы сделали это, добавив соответствующие данные в Таблицу 3, и благодарим рецензента за это хорошее предложение.
Насколько большое RSD может повлиять на результаты, полученные с помощью используемого алгоритма?
Мы не обнаружили выраженного влияния отклонения сигнала и подчеркнули это в исправленной версии рукописи.
- Рисунок 4 некачественный. Пожалуйста, увеличьте шрифты на осях.
Рецензент совершенно прав. Мы увеличили цифры, чтобы улучшить читаемость.

Reviewer 2 Report
The manuscript discusses the possibility of detecting A and B influenza viruses by surface-enhanced Raman scattering spectroscopy. The authors achieve 93% accuracy in differentiation of such viruses by using machine learning technology. The manuscript is generally well written. Its results are significant and interesting for specialists in the related areas. My only remark concerns the introduction. Its logic is not very convincing. It looks like the technologies and algorithms for the automatic recognition of viruses have been worked out in many examples, and the only unsolved problem is how to apply those conventional technologies for the detection of A and B influenza viruses. Therefore, the novelty of this work should be better highlighted in the introduction.
Author Response
We would like to thank all the reviewers for giving their time and energy in reviewing our manuscript. We believe that their critiques have greatly improved the quality and accessibility of our paper. It is now much more self-contained regarding the approach as a lot more details and references has been included in the revised manuscript.
Reviewer 2
The manuscript discusses the possibility of detecting A and B influenza viruses by surface-enhanced Raman scattering spectroscopy. The authors achieve 93% accuracy in differentiation of such viruses by using machine learning technology. The manuscript is generally well written. Its results are significant and interesting for specialists in the related areas.
We thank the reviewer for their kind words.
My only remark concerns the introduction. Its logic is not very convincing. It looks like the technologies and algorithms for the automatic recognition of viruses have been worked out in many examples, and the only unsolved problem is how to apply those conventional technologies for the detection of A and B influenza viruses. Therefore, the novelty of this work should be better highlighted in the introduction.
We accept the reviewer's remark which was also emphasized by Reviewer1. Please see the answer to their comment.

Round 2
Reviewer 1 Report
The authors responded thoroughly to all my questions. I have no further comments and recommend the manuscript for publication in the Biosensors journal.